# Intermittent preventive treatment with Sulphadoxine-Pyrimethamine (IPTp-SP) is associated with protection against sub-microscopic *P. falciparum* infection in pregnant women during the low transmission dry season in southwestern Cameroon: A Semi - longitudinal study

Tobias O. Apinjoh[1,2]*, Vincent N. Ntui[1], Hanesh F. Chi[3], Marcel N. Moyeh[1,2], Cabrel T. Toussi[1], Joel M. Mayaba[1], Livinus N. Tangi[3], Pilate N. Kwi[1], Judith K. Anchang-Kimbi[4], Jodie Dionne-Odom[5], Alan T. N. Tita[6], Eric A. Achidi[1], Alfred Amambua-Ngwa[7], Vincent P. K. Titanji[1]

1 Department of Biochemistry and Molecular Biology, University of Buea, Buea, Cameroon, 2 Department of Chemical and Biological Engineering, The University of Bamenda, Bambili, Cameroon, 3 Department of Microbiology and Parasitology, University of Buea, Buea, Cameroon, 4 Department of Zoology and Animal Physiology, University of Buea, Buea, Cameroon, 5 Division of Infectious Diseases, Department of Medicine, University of Alabama at Birmingham, Birmingham, AL, United States of America, 6 Division of Maternal-Fetal Medicine, Department of Obstetrics and Gynecology and Center of Women's Reproductive Health, University of Alabama at Birmingham, Birmingham, AL, United States of America, 7 Medical Research Council Unit The Gambia at London School of Hygiene and Tropical Medicine, Fajara, Banjul, The Gambia

* apinjoh.tobias@ubuea.cm

## Abstract

The current guidelines for malaria prevention and control during pregnancy in Africa is predicated on the prevention of infection and/or disease through intermittent preventive treatment in pregnancy (IPTp), insecticide-treated nets (ITNs) and effective malaria case diagnosis and management. Concerns that increasing SP resistance in some areas of SSA may have compromised IPTp-SP efficacy prompted this contemporaneous study, designed to assess the prevalence and risk factors of sub-microscopic infection in parturient women during the low transmission season in Mutengene, a rapidly growing semi-urban area in Southwest Region, Cameroon. Pregnant women originally reporting for the establishment of antenatal clinic care during the dry season were followed-up to term and their pregnancy outcomes recorded. About 2 ml of venous blood was collected for malaria diagnosis using *PfHRP2*/pLDH malaria rapid diagnostic kit and light microscopy. DNA was extracted from dried blood spots by the Chelex-100 method and the *Plasmodium falciparum* status detected by nested PCR amplification of the 18SrRNA gene using specific predesigned primers. Of the 300 women enrolled, the proportion of malaria parasite infected as determined by microscopy, RDT and PCR was 12.9%, 16.4% and 29.4% respectively, with 39.9% overall infected with *P. falciparum* by microscopy and/or RDT and/or PCR and a very low-density infection,

**Data Availability Statement:** All relevant data are within the paper.

**Funding:** The study received received financial support in the form of a grant from the Developing Excellence in Leadership and Genetics Training for Malaria Elimination in sub-Saharan Africa (DELGEME) program (Grant 107740/Z/15/Z) sponsored by the Developing Excellence in Leadership, Training, and Science (DELTAS) Africa initiative and the H3Africa through the Alliance for the African Academy of Science – Grant n0. H3AFull/17/008. The funders had no role in study design, data collection and analysis, decision to publish, or preparation of the manuscript.

**Competing interests:** The authors have declared that no competing interests exist.

averaging 271 parasites per microliter of blood. About 25.0% (68/272) of women who were negative by microscopy were positive by PCR (submicroscopic *P. falciparum* infection), with primigravidae and IPTp-SP non usage identified as independent risk factors for submicroscopic *P. falciparum* parasitaemia while fever history (aOR = 4.83, 95% CI = 1.28–18.22, p = 0.020) was associated with risk of malaria parasite infection overall. IPTp-SP use (p = 0.007) and dosage (p = 0.005) significantly influenced whether or not the participant will be malaria parasite negative or carry submicroscopic or microscopic infection. Although Infant birthweight and APGAR score were independent of the mother's *P. falciparum* infection and submicroscopic status, infant's birthweight varied with the gravidity status (p = 0.001) of the mother, with significantly lower birthweight neonates born to primigravidae compared to secundigravidae (p = 0.001) and multigravidae (p = 0.003). Even in holo-endemic dry season, there exists a large proportion of pregnant women with very low density parasitaemia. IPTp-SP seems to be relevant in controlling submicroscopic *P. falciparum* infections, which remains common in pregnant women, and are hard to diagnose, with potentially deleterious consequences for maternal and fetal health. Future studies should be carried out in hyper-endemic malaria foci where the parasitemia levels are substantially higher in order to confirm the efficacy of IPTp-SP.

## Introduction

Malaria still remains a significant health problem especially in sub-Saharan Africa in spite of the tremendous control and elimination efforts that decreased cases and deaths over the last decade. Pregnant women constitute at major risk group, with some 12 million pregnancies per year in the WHO Africa region exposed to *Plasmodium spp* infection during gestation. Malaria in pregnancy (MiP) is known to adversely affect both the mother and her progeny [1], with falciparum infection and disease, commonly manifested as maternal anaemia, pre-term delivery and low birth weight [2–4] linked to poor infant development, increased neonatal and infant morbidity and mortality especially in low transmission settings [1, 5]. Expectant mothers are also at higher risk for severe malaria, pregnancy loss and death compared to their non-pregnant peers, with a mortality rate up to 50% for severe infections [6].

Current guidelines for malaria prevention and control during pregnancy in Africa emphasize on the prevention of infection and/or disease through intermittent preventive treatment (IPTp), the use of insecticide-treated nets (ITNs) and prompt and effective malaria case management [1]. Pregnancy poses a unique challenge for *falciparum* malaria diagnosis, as parasites sequester in the placenta and the circulating parasitaemia is very low and therefore undetectable by peripheral blood smear microscopy or rapid diagnostic test [7–10]. Microscopically detectable *P. falciparum* parasitaemia in peripheral blood is, therefore, a poor indicator of the actual state of infection in pregnancy, with significant adverse implications for the mother, developing fetus and infant [1] of unrecognized malaria infected parturient subjects.

Over the past decades, several studies have demonstrated submicroscopic *P. falciparum* infection, detectable only by sensitive molecular (PCR) methods in Sub-Saharan Africa (SSA) and documented high prevalence rates of low-density parasitaemia, shedding new light on the real prevalence of malaria infections [11], especially in pregnant women [2]. Such submicroscopic infections may negatively affect health and constitute reservoirs for continued disease

transmission [10] since they are usually asymptomatic and therefore remain untreated during pregnancy.

Although the consequences of malaria in pregnancy (MiP) on the mother and her progeny are known, information on the effectiveness of intervention strategies in limiting the disease morbidity and/or mortality in some endemic settings remain limited. Several studies have reported the effectiveness of IPTp with sulphadoxine-pyrmethamine (SP) (IPTp-SP) in the prevention of pregnancy associated malaria in areas with moderate to high transmission but not so much in areas with substantial reduction of malaria transmission [12]. Concerns that increasing SP resistance in SSA [13] may have compromised the efficacy of this chemoprophylactic control measure [14], makes WHO's call for enhanced regular monitoring of IPTp effectiveness ever more relevant in all malaria-endemic ecosystems.

Previous MIP studies in the southwest region of Cameroon have documented IPTp-SP coverage [15, 16] as well as its effectiveness on adverse pregnancy outcomes [16] and malaria parasitaemia [15]. Nonetheless, all assessments of malaria parasite infection, relevant for the management of the disease, accrue mainly to light microscopy, with currently no data on the prevalence of submicroscopic infection and any perinatal/prenatal risk factors in the context of IPTp-SP. In this area, there has been the intensification of malaria control through the distribution of free long-lasting insecticide treated nets in the last decade to disrupt vector dynamics and reduce disease transmission. However, this may have been interjected by socio-political instability since 2016 that can potentially exacerbate the malaria situation as reported elsewhere [17]. The aims of the present investigation were to determine the prevalence of submicroscopic malaria parasite infection in the context of IPTp-SP and its relationship to pregnancy outcomes, as well as to identify any additional risk factors of malaria in pregnancy in parturient women attending antenatal enrolment mainly during the holo-endemic dry season.

## Materials and methods

### Ethical clearance

Ethical clearance (Ref. N°2019/09/1188/CE/CNERSH/SP) was obtained from the Comite National D'ethique de la Recherche pour la Sante Humaine (CNERSH) while administrative authorization was obtained from the South West Regional delegation of Public Health, District Medical Officer and Chief Medical Officers in charge of the district and health facilities respectively. Upon approval, participants (or their guardians) who signed a written informed consent after adequate sensitization about the study objectives, risks and possible benefits were enrolled. Assent was equally obtained from the guidance of pregnant women below 16 years of age in line with national gynecological guidelines.

### Study area

This study was carried out in Mutengene (4˚4'30" N 9˚21'36" E) a semi-urban setting located at approximately 220 m above sea level in the Tiko Health District, located at the foot of Mount Cameroon in the South West Region of Cameroon. The population of the area is highly heterogeneous, consisting of individuals from almost all the ethnic groups of Cameroon and some parts of neighboring Nigeria [15]. The area has an equatorial climate characterized by daily temperatures ranging between 20–33˚C, average annual rainfall of 2625 mm, relatively high humidity and precipitation and two major seasons; a long rainy season (March–November) and a short dry season (November–February) [18]. *Plasmodium spp* transmission is perennial in the area, with *Anopheles gambiae* being the dominant vector species and *P. falciparum* accounting for most malaria infections [19, 20]. Intermittent preventive treatment in pregnancy with SP from 4 months was implemented in Mutengene since 2006, in line with the

World Health Organization guidelines for the delivery of IPTp at each scheduled visit after "quickening" (16 weeks) to ensure that a high proportion of women receive a minimum of 2 doses of IPT [15].

## Study design, population and sampling

This was a prospective cohort study carried out in the Mutengene Integrated Health Centre between October 2019 to March 2020. Pregnant women resident in Mutengene and surrounding villages reporting for antenatal clinic in the outpatient departments were eligible for enrolment (Fig 1). Individuals who did not consent or were unable to provide samples were excluded. A structured questionnaire was used to document demographic and clinical data, followed by venous blood collection (about 2 ml) from the same patient into labelled tubes containing EDTA. All samples were stored between 2 to 8°C on ice blocks until analysis. The malaria parasite status in blood was immediately checked using mRDT, blood smears prepared on grease-free slides as described [21] and about 50 μl blood spotted on labeled Whatman No. 3 filter paper and allowed to air dry for further molecular analysis.

## Data collection and definition of terms

A structured questionnaire was used to document demographic and clinical information including maternal age, last menstrual period, gravidity, parity, gestational age, history of fever

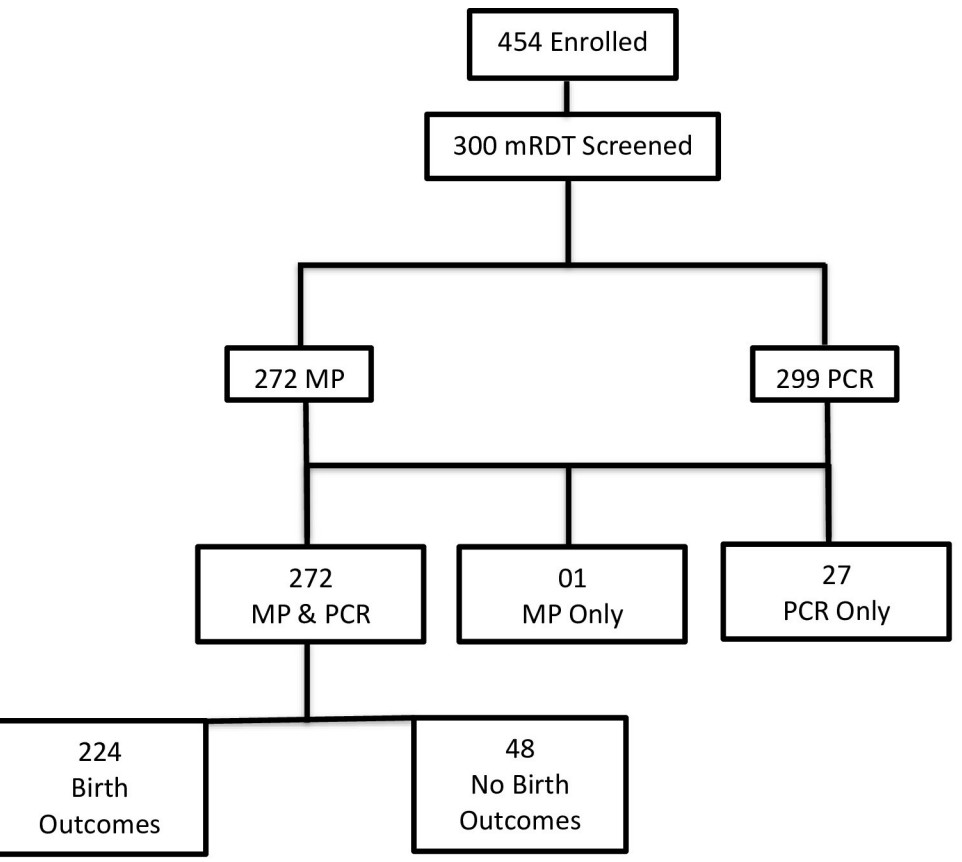

**Fig 1. Flow chart of participant enrolment and analysis.** MP (Malaria parasitemia), mRDT (malaria Rapid Diagnostic Test), PCR (Polymerase chain reaction).

attack during gestation as well as infant birthweight and Apgar score at enrollment and/or delivery from the participant, ANC cards, participant's medical record book, and health center maternal care register. The participant's axillary temperature was measured using a digital thermometer, with current fever defined as body temperature ≥37.5˚C, with febrile individuals defined as participants with current fever or history of fever at enrolment. Insecticide-treated nets use was defined as reportedly sleeping under a bed net or ITN the previous night. Ethnicity was defined as the self-reported ethnic group of the mother (or the father if the mother's could not be obtained) [22]. Low birthweight was defined as infant birthweight below 2.5 kg and Prematurity as baby born before 37 weeks of gestation.

Venous blood (about 2 ml) was collected from the same patient into labelled tubes containing EDTA and stored between 2 to 8˚C on ice blocks until analysis. The malaria parasite status in blood was immediately checked using mRDT, blood smears prepared on grease-free slides as described [21] and about 50 μl blood spotted on labeled Whatman No. 3 filter paper and allowed to air dry overnight. The dried blood spots were packaged on separate ziplock bags with desiccants and stored in a cold room prior to DNA extraction for further molecular analysis. *Plasmodium falciparum* infections status was classified as negative if all tests (RDT, Microscopy and PCR, when applicable) were negative; submicroscopic if Microscopy (and RDT, when applicable) was negative but the PCR was positive and microscopic if microscopy or RDT (when applicable) was positive whether the PCR was positive or negative.

**IPTp-SP usage and dosage.**   Information on the previous use of IPTp-SP and dosage was documented at enrollment from the participant, ANC cards, her medical record book, and health center maternal care register. Subsequent IPTp-SP usage and dosage was confirmed at each scheduled visit by directly observed administration of the drug by nurses/midwives in line with national guidelines.

## Laboratory analysis

**Haemoglobin level.**   Hemoglobin concentrations at enrolment were determined using a hemoglobinometer (Hangzhou Sejoy Electronics, Hangzhou, China) and anemia defined as Hb <11.0 g/dL [23].

**Malaria rapid diagnostic test.**   The presence of *Plasmodium spp* in blood was assessed using the *PfHRP2*/pLDH malaria rapid diagnostic kit (SD Bioline™, Alere, South Korea) according to the manufacturer's instructions. Briefly, 5 μl of whole blood sample from each tube was placed in the sample well of the RDT cassette and three drops of diluent added to the buffer well. The results were then read after 15 minutes, the presence of two (or three), one or no distinct line indicative of a positive, negative or invalid result respectively.

**Light microscopy.**   Thin and thick blood smears were prepared on labeled grease free slides and allowed to air dry. The thin films were fixed with methanol and both smears stained by submerging in 10% Giemsa for 15 minutes as described [21]. Subsequently, the stained slides were then air-dried and viewed under the 100X oil immersion objective of a binocular Olympus microscope. Each slide was examined by two independent microscopists and was considered negative if no parasite was seen after observing 200 high power fields. With each positive slide, the parasite densities were calculated by counting the parasites against a minimum of 200 leucocytes and assuming an average leucocyte count of 8000 per μl of blood [24].

**DNA extraction and PCR.**   DNA was extracted from dried blood on Whatman No. 3 filter paper using the Chelex-100 method as previously described [25], with some modifications. Three punches of each filter paper sample was added into an Eppendorf tube containing 1ml of 0.5% saponin in 1xPBS and incubated overnight at 4˚C to lyse RBCs. The punches were then washed in 1ml of 1% PBS at 4˚C for 30minutes and then transferred into a newly labeled

tube containing 50μl of 20% Chelex and 150 μl distilled water on a thermomixer. The tube was then incubated at 99˚C for 10 minutes to elute the DNA, vortexed for about 2 minutes and then centrifuged at 11000 x g for 2 minutes. The supernatant/presumed DNA extract was then transferred into a new Eppendorf tube and the stocked stored at -20˚C for long term use. An aliquot (bout 50ul) of the DNA sample was kept at 4˚C as working stock for immediate use. The *P. falciparum* 18S rRNA gene was then amplified by PCR as described previously [26] using specific predesigned primers (Table 1).

All PCR amplifications were carried out on a thermal cycler (BioRad, CA, USA), in a total volume 15 μl, consisting of 7.5 μl of 2x Taq master mix and 0.25 μl each of forward (rPLU5) and reverse (rPLU6) primers. In the primary PCR, 5.0μl of nuclease free water and 2 μl of extracted genomic DNA was added to the PCR tube and the reaction carried out under the following cycling conditions: Initial denaturation at 95˚C for 3 minutes and 25 cycles of denaturation at 94˚C for 30 seconds, annealing at 55˚C for 1 min, extension at 68˚C for 2 minutes and final extension at 68˚C for 3 minutes. In the Nested PCR, 6.0 μl of nuclease free water and 1 μl of primary PCR product was added to the PCR tube containing the master mix and primers, and the reaction carried out under the same cycling conditions as the primary PCR, except that the annealing temperature was modified to 61˚C and the number of cycles to 30.

The nested PCR amplified fragments were separated in 2.5% ethidium bromide-stained agarose gel electrophoresis in 0.5X TAE buffer using a power pack (Bio-rad, CA, USA) at 100V for 15 minutes using relative to 100bp molecular weight maker and observed using UV transilluminator and a Gel Documentation System (Molecular Imager® Gel Doc™ XR+ System with Image Lab™ Software, Bio-Rad, Berkeley California, USA). A sample was considered positive if a 205 bp band was detected. *P. falciparum* 3D7 and distilled water served as positive and negative control respectively in every set of reactions.

## Data analysis

Data obtained from the study was analyzed using SPSS Statistics 20 (SPSS Inc, Chicago USA). Qualitative variables were compared using the Chi Square test while differences in group means were assessed using the student's t test or analysis of variance. Association analysis of perinatal and prenatal risk factors on maternal susceptibility to submicroscopic or microscopic *P. falciparum* infection was undertaken by logistic regression; only variables of known clinical relevance or with p-value $< 0.25$ in univariate analysis were included in multivariate models. *Plasmodium falciparum* infections status was classified as negative if all tests (RDT, Microscopy and PCR, when applicable) were negative; submicroscopic if Microscopy (and RDT, when applicable) was negative but the PCR was positive and microscopic if microscopy or RDT (when applicable) was positive whether the PCR was positive or negative. Individuals who tested negative as defined above were included as the comparison group. A probability (P) value $<0.05$ was considered statistically significant at a 95% confidence interval.

**Table 1. PCR primers for *Plasmodium falciparum* detection.**

| Gene | Primers | Sequence | band size (bp) |
|------|---------|----------|----------------|
| **Primary PCR** | | | |
| Mitochondrial *coxI* gene | rPLU5 | (5′-GAC CTG CAT GAA AGA TG-3′) | 1200 |
| | rPLU6 | (5′-GTA TCG CTT TAA TAG GCG-3′) | |
| **Nested PCR** | | | |
| 18S rRNA | fal1 | (5′-GGAATGTTATTGCTAACAC-3′) | 205 |
| | fal2 | (5′-AAT GAA GAG CTG TGT ATC-3′) | |

## Results

A total of 454 pregnant women aged 14–40 years were originally enrolled into the study during the dry season (Fig 1). The first 300 women sequentially enrolled were screened by rapid diagnostic tests, microscopy and PCR and used for all subsequent analyses. Two hundred and seventy-two women had results for both PCR and light microscopy while 01 and 27 had results only for microscopy and PCR respectively. Two hundred and twenty-four women were successfully followed-up to term and their pregnancy outcomes obtained.

### Demographic and obstetric characteristics of the study population

The baseline socio-demographic and obstetric characteristics of these parturient women and the infants born from this pregnancy are summarized in Table 2. The majority of the expectant mothers came from Mutengene and surrounding areas (92.3%), the Semi-Bantu ethnic group (75.3%) and were bed net users (65.4%), anemic (60.8%), more than 25 years old (51.2%), in the third trimester of gestation (50.8%) and multigravida (44.9%) at enrolment. While a few women had taken up to 4 IPTp-SP doses prior to ANC, only 35.5% (81/228) of the women beyond 18 weeks of gestation reportedly took at least a dose of SP, even though most women were in third trimester of gestation at enrolment. More than three quarters (78.9%, 235/298) of the women had taken IPTp-SP at term, with more than half (55.7%, 166/298) of them on at least 3 prophylactic doses. The average gestational age at delivery was 39.4 weeks, although 3.1% and 3.6% neonates were low birthweight or premature respectively.

### Prevalence of malaria parasite infection as a function of diagnostic method

The proportion of malaria parasite infected women as determined by microscopy, RDT and PCR was 12.9% (35/272), 16.4% (49/299) and 29.4% (88/299) respectively, with 39.9% (110/276) overall infected with *Plasmodium falciparum* by microscopy and/or RDT and/or PCR (Fig 2A). While PCR was the most sensitive screening technique, detecting more than twice the number of infected individuals, compared to light microscopy or RDT, the latter performed better than the former, the gold standard method for malaria diagnosis. Malaria parasite infected parturient women in the cohort harbored very low-density infection, averaging 271 parasites per microliter of blood. Almost 2 out of every 5 (38.4%) women who reported no history of fever at enrolment, tested positive for *P. falciparum*, while (26%, 19/73) women on IPTp-SP tested positive, although the prevalence of malaria parasite infection was higher (p = 0.005) in IPTp-SP non-users compared to those who had taken at least a dose of SP.

### Submicroscopic *P. falciparum* infection

One out of every four (25.0%, 68/272) women who were negative by microscopy were positive by PCR (submicroscopic *P. falciparum* infection), twice the proportion of cases detected in peripheral blood smears (12.9%, 35/272) at enrolment (Fig 2B). The prevalence of submicroscopic infection was independent of all potential perinatal and prenatal risk factors assessed in this study (Table 3), although those who had taken at least a dose of IPTp-SP tended to harbor less submicroscopic *P. falciparum* infections (p = 0.064) compared to their counterparts who were not on IPTp-SP. Gravidity and IPTp-SP usage were identified as independent risk factors for submicroscopic *P. falciparum* parasitaemia following multivariate analysis, with primigravidae more susceptible (aOR = 2.92, 95% CI = 1.19–7.18, p = 0.019) and IPTp-SP users at lower risk (aOR = 0.12, 95% CI = 0.01–0.98, p = 0.048), after adjusting for covariates (Table 3).

**Table 2. Baseline socio-demographic and obstetric characteristics of parturient mothers and their Infants from Mutengene, Southwest, Cameroon.**

| Parameter | Sub-class | Total No. | Mean ± SD [Range] or % (n) |
|---|---|---|---|
| **Maternal characteristics, mean ± SD [Range]** | | | |
| Age (years) | | 298 | 26.2 ± 6.0 [14–40] |
| Gestational Age (weeks) | enrolment \| term | 296 \| 220 | 25.1 ± 7.2 [7–40] \| |
| | | | 39.4 ± 1.6 [29–44] |
| IPTp-SP Dosage | enrolment \| term | 228 \| 298 | 0.8 ± 1.3 [0–4] \| |
| | | | 2.3 ± 1.5 [0–4] |
| Weight (kg) | | 288 | 69.7 ± 12.7 [40–120] |
| Hb (g/dl) | | 288 | 10.8 ± 1.4 [7.7–21.4] |
| GMPD$^\$$ (parasites /μl blood) | | 57 | 271 [78–36,178] |
| **Maternal characteristics, % (n)** | | | |
| Age group (years) | ≤ 20 | 298 | 22.1 (66) |
| | 21–25 | | 26.4 (79) |
| | > 25 | | 51.2 (153) |
| Gravidity | Primigravidae | 296 | 30.1 (89) |
| | Secundigravidae | | 25.0 (74) |
| | Multigravidae | | 44.9 (133) |
| Parity | 0 | 296 | 32.8 (97) |
| | 1–3 | | 59.1 (249) |
| | ≥ 4 | | 7.4 (31) |
| Trimester of gestation | First | 297 | 2.7 (8) |
| | Second | | 46.5 (138) |
| | Third | | 50.8 (151) |
| Ethnicity | Bantu | 299 | 20.7 (62) |
| | Semi-bantu | | 75.3 (225) |
| Residence | Mutengene & environs | 297 | 92.3 (274) |
| | Douala | | 7.7 (23) |
| Fever at enrolment | History of fever | 299 | 5.4 (16) |
| | Axillary temperature ≥ 37.5˚C | 298 | 16.1 (48) |
| | Febrile | 298 | 19.5 (58) |
| IPTp-SP Use | Enrolment (beyond 18 weeks) \| term | 228 \| 298 | 35.5 (81) \| 78.9 (235) |
| IPTp-SP Dosage | 0 | | 64.5 (147) \| 21.1 (63) |
| | 1 | | 9.2 (21) \| 7.7 (23) |
| | 2 | | 9.6 (22) \| 15.4 (46) |
| | ≥ 3 | | 16.7 (38) \| 55.7 (166) |
| Bednet Use | | 298 | 65.4 (195) |
| Bednet & IpTp-SP Use | Enrolment (beyond 18 weeks) \| term | 228 \| 297 | 28.9 (66) \| 51.9 (154) |
| Maternal anaemia | | 288 | 60.8 (175) |
| **Infant characteristics, mean ± SD [Range]** | | | |
| Birthweight (kg) | | 224 | 3.3 ± 0.4 [2.2–4.5] |
| APGAR (5 mins) | | 219 | 8.5 ± 0.9 [5.0–10.0] |
| **Infant characteristics, % (n)** | | | |
| Low birthweight | | 224 | 3.1 (7)[&] |
| Prematurity | | 220 | 3.6 (8)[#] |

$^\$$GMPD = Geometric Mean Parasitaemia Density;

[&]Low birthweight = birthweight < 2.5 kg;

[#]Prematurity = baby born before 37 weeks.

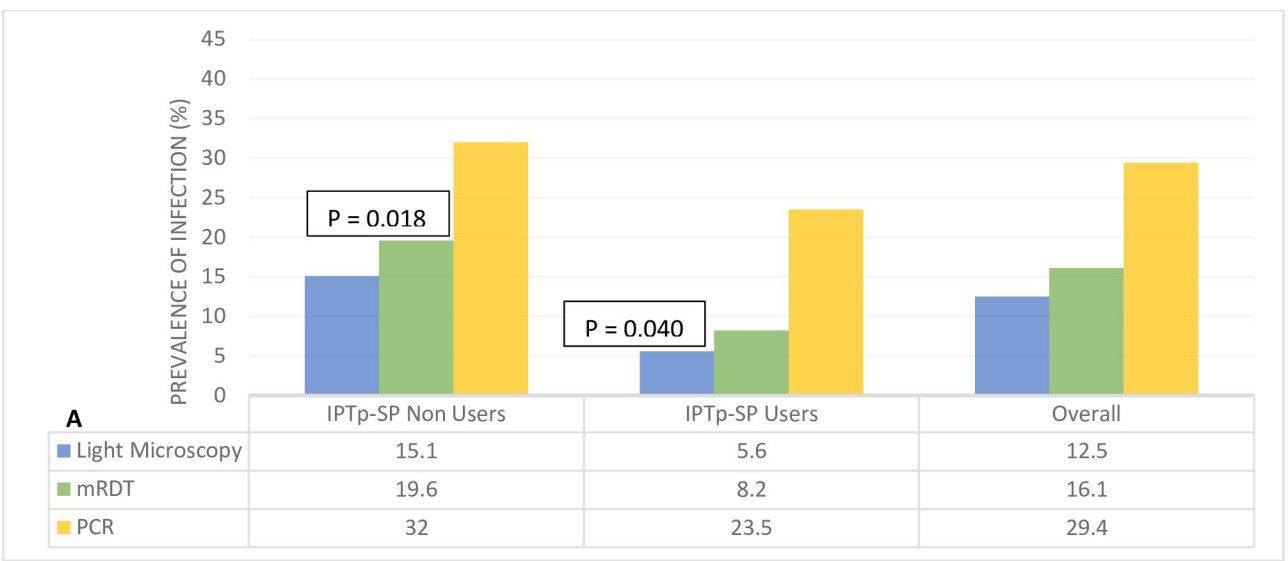

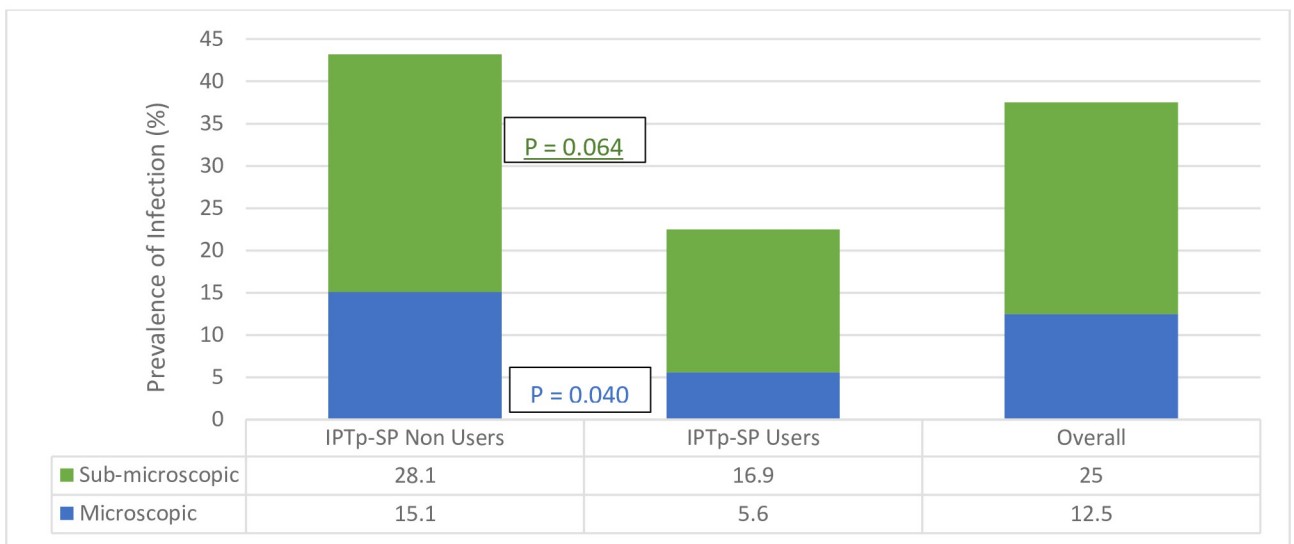

**Fig 2. Prevalence of malaria parasite infection in pregnant women reporting for ANC in the dry season in southwestern Cameroon.**

### Risk factors for submicroscopic and microscopic malaria parasite infection

Following univariate analysis, the overall prevalence of malaria parasite (submicroscopic and microscopic) infection was associated with IPTp-SP usage (p = 0.005), cumulative number of IPTp doses (p = 0.022), reported history of fever (p = 0.027) and trimester of gestation (p = 0.028) but independent of age group (p = 0.723), gravidity (p = 0.494), parity (p = 0.282), ethnicity (p = 0.200), anaemia status (p = 0.254), town of residence and bednet usage (p = 0.471). In the final multivariate model, fever history remained the only statistically significant factor contributing to malaria parasite infection status, with women with a history of fever in the last 48 hours more susceptible (aOR = 4.83, 95% CI = 1.28–18.22, p = 0.020) to infection, after adjusting for covariates.

**Table 3. Relationship between perinatal and prenatal risk factors and maternal susceptibility to submicroscopic *P. falciparum* infection at enrolment.**

| Variable | Sub-class | Submicroscopic *P. falciparum* infection | | | Adjusted P value |
|---|---|---|---|---|---|
| | | % (n) | Unadjusted | OR [95%CI] | |
| Age group (years) | ≤ 20 | 28.1 (18) | $X^2 = 0.427$; p = 0.808 | 0.48 [0.18–1.29] | 0.146 |
| | 21–25 | 23.6 (17) | | 0.69 [0.30–1.56] | 0.366 |
| | > 25 | 24.4 (33) | | Ref | |
| Gravidity | Primigravidae | 30.1 (25) | $X^2 = 2.103$; p = 0.349 | 2.92 [1.19–7.18] | **0.019** |
| | Secundigravidae | 26.6 (17) | | 2.07 [0.90–4.76] | 0.070 |
| | Multigravidae | 21.3 (26) | | Ref | |
| Trimester of gestation | First | 37.5 (3) | $X^2 = 1.738$; p = 0.419 | 0.38 [0.04–4.01] | 0.421 |
| | Second | 27.5 (36) | | 0.83 [0.39–1.79] | 0.620 |
| | Third | 22.0 (29) | | Ref | |
| Ethnicity | Semi-bantu | 23.0 (47) | $X^2 = 1.737$; p = 0.187 | 0.59 [0.28–1.24] | 0.162 |
| | Bantu | 31.6 (18) | | Ref | |
| History of fever | Yes | 24.1 (63) | $X^2 = 2.558$; p = 0.110 | 2.86 [0.74–11.11] | 0.129 |
| | No | 45.5 (5) | | Ref | |
| IPTp-SP Use | Yes | 16.9 (12) | $X^2 = 3.437$; p = 0.064 | 0.12 [0.01–0.98] | **0.048** |
| | No | 28.1 (52) | | Ref | |
| IPTp-SP Dosage | 0 | 28.1 (52) | $X^2 = 4.743$; p = 0.093 | 0.18 [0.02–1.73] | 0.138 |
| | 1–2 | 11.1 (4) | | 0.68 [0.15–3.05] | 0.611 |
| | ≥ 3 | 22.9 (8) | | Ref | |
| Bednet Use | Yes | 22.7 (22) | $X^2 = 0.468$; p = 0.494 | 1.51 [0.77–2.97] | 0.232 |
| | No | 26.4 (46) | | Ref | |
| Maternal anaemia status | Anaemic | 25.5 (40) | $X^2 = 0.094$; p = 0.759 | 0.85 [0.44–1.63] | 0.622 |
| | Non anaemic | 23.8 (25) | | Ref | |

Values in parentheses denote total participants with valid values for this variable; Bold text indicate significant P values.

## IPTp-SP and malaria parasite infection

The prevalence of malaria parasite infections by mRDT (p = 0.018) and light microscopy (p = 0.040) as well as overall (p = 0.005) was significantly lower in IPTp-SP users when compared to non-users, with the latter also tending (p = 0.064) to harbor more sub-microscopic *P. falciparum* infections (Fig 2). There was, however, no significant difference (p = 0.200) in the prevalence of *P. falciparum* infection in IPTp-SP users and non-users as determined by PCR. There was a significant association (p = 0.001) between the malaria parasite status by microscopy and IPTp-SP dosage, with none of the women infected after taking at least 2 doses of IPTp-SP. Nevertheless, there was no association between infection status by mRDT (p = 0.986) and PCR (p = 0.217) as well as the overall parasitaemia (p = 0.334) and submicroscopic infection status (p = 0.306) and the dosage of SP taken. Additionally, IPTp-SP use (p = 0.007) and dosage (p = 0.005) significantly influenced whether or not the participant will be malaria parasite negative or carry submicroscopic or microscopic infection (Fig 3). While the proportion of submicroscopic and microscopic infection was significantly reduced in IPTp-SP users compared to non-users, none of the women harboring microscopic infection after at least two doses of the IPTp-SP (Fig 3).

## Risk factors and birth outcomes

The mean birthweight and APGAR score of the 193 and 187 neonates duly assessed was 3.3 ± 0.4 kg and 9.8 ± 0.9 respectively. Malaria parasite negative women had similar

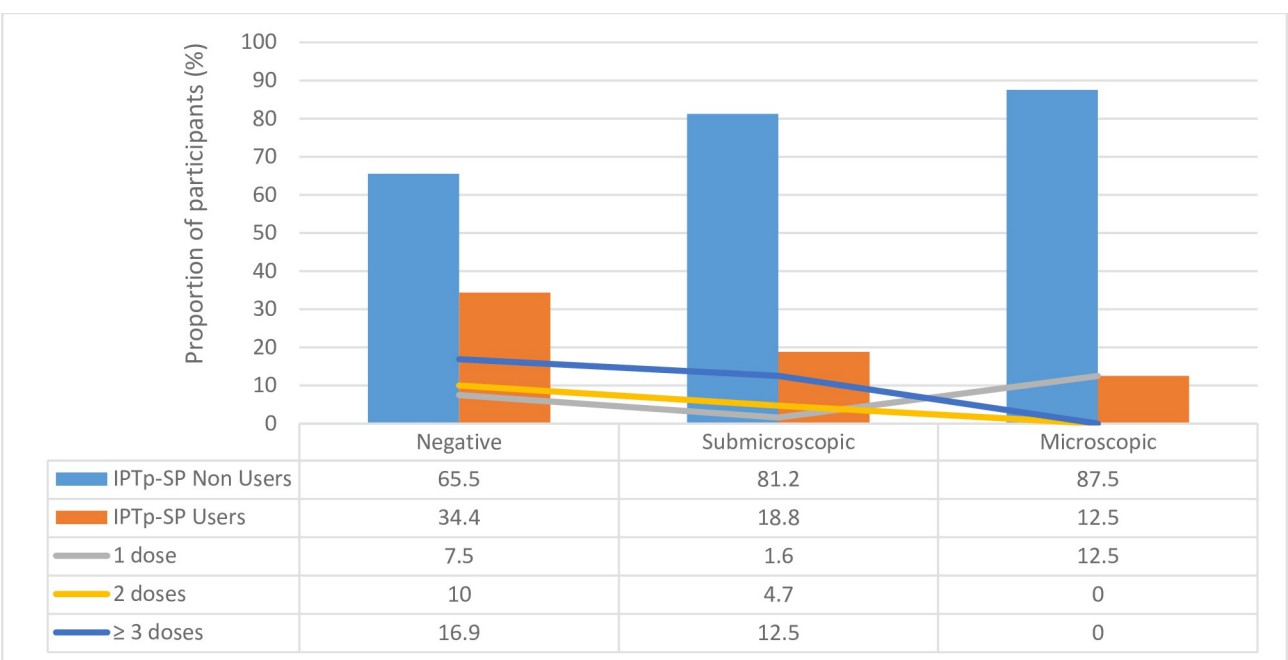

**Fig 3. Influence of IPTp-SP usage and dosage on the malaria parasite infection status of pregnant women reporting for ANC in the dry season in southwestern Cameroon.**

birthweights (p = 0.158) and APGAR scores (p = 0.672) when compared to those with submicroscopic or microscopic infection. APGAR score did not vary with any of the risk factors assessed. However, infant's birthweight varied with the gravidity status (p = 0.001) of the

**Table 4. Relationship between perinatal and prenatal risk factors and infant birthweight and APGAR score.**

| Variable | Sub-class | Mean ± SD (n) | |
|---|---|---|---|
| | | **Birthweight (kg)** | **APGAR** |
| Gravidity | Primigravidae | 3.13 ± 0.39 (65)[a] | 8.42 ± 0.81 (64) |
| | Secundigravidae | 3.39 ± 0.41 (55) | 8.52 ± 0.95 (54) |
| | Multigravidae | 3.34 ± 0.41 (103) | 8.44 ± 0.84 (100) |
| P value | | **0.001** | 0.812 |
| Ethnicity | Semi-bantu | 3.32 ± 0.40 (167) | 8.47 ± 0.90 (163) |
| | Bantu | 3.19 ± 0.45 (47) | 8.46 ± 0.78 (46) |
| P value | | 0.075 | 0.947 |
| IPTp-SP Use (at term) | Yes | 3.27 ± 0.41 (195) | 8.42 ± 0.85 (192) |
| | No | 3.40 ± 0.42 (29) | 8.67 ± 0.92 (27) |
| P value | | 0.109 | 0.175 |
| IPTp-SP Dosage (at term) | 1 | 3.11 ± 0.71 (11) | 7.89 ± 0.60 (9) |
| | 2 | 3.37 ± 0.45 (40) | 8.52 ± 0.82 (40) |
| | $\geq$ 3 | 3.26 ± 0.37 (149) | 8.45 ± 0.86 (148) |
| P value | | 0.128 | 0.121 |
| Bednet Use | Yes | 3.28 ± 0.41 (149) | 8.46 ± 0.80 (145) |
| | No | 3.31 ± 0.44 (74) | 8.44 ± 0.97 (73) |
| P value | | 0.651 | 0.848 |

[a]Significantly lower than the corresponding values in secundigravidae (p = 0.008) and multigravidae (p = 0.005).

mother (Table 4), with significantly lower birthweight neonates born to primigravidae compared to secundigravidae (p = 0.001) and multigravidae (p = 0.003). Similarly, babies born to nulliparous women (3.14 ± 0.39) had significantly lower birthweights when compared to their counterparts with 1–3 live births (3.34 ± 0.42, p = 0.003) and at least 4 children (3.53 ± 0.31, p = 0.002) counterparts. Apart from the significant association (p = 0.012) between low birthweight and gravidity, with higher prevalence in primigravidae, the prevalence of prematurity and low birthweight was also independent the mother's *P. falciparum* infection, submicroscopic status and all other risk factors.

## Discussion

This study responds to WHO's call for enhanced regular monitoring of IPTp effectiveness and documents, for the first time, significant sub-microscopic *P. falciparum* infection in parturient women attending antenatal enrolment during the low transmission dry season in southwestern Cameroon, demonstrating that SP is highly effective in controlling malaria parasite infection in pregnancy and thereby preventing negative outcomes such as low birthweight, preterm delivery and stillbirth. The present study is undertaken in a multi-ethnic and urban town, Mutengene, which is representative of a rapidly growing semi-urban area where more than 50% of Cameroonians live. Results obtained here will be reflective of similar locations in the same ecosystem. Given the potential of SP resistance, the study is important as it demonstrates that SP, a well-tolerated drug is still highly effective against *P. falciparum* in the area under study.

The overall 39.9% prevalence of *Plasmodium falciparum* infection in this study mirrors previous reports [6] and shows that malaria in pregnancy is still an important health issue in the area during this period in spite of the reduced transmission. This can be explained by the low IPTp-SP uptake recorded in this study, since the intervention has been shown to substantially reduce the overall prevalence of malaria and contribute to the reduction in placental malaria and deleterious birth outcomes that follows from peripheral malaria [12]. However, the fact that up to 26% of women reportedly under IPTp-SP were parasitaemic suggests that the malaria prophylaxis might not be highly effective [27] and more effective preventive measures are needed [28].

The fact that PCR detected up to twice the proportion of cases identified in peripheral blood smears at enrolment, demonstrates that molecular techniques remain more sensitive [9] and the challenge of detecting malaria in pregnancy with the routine clinical diagnostic tools. Previous studies in Cameroon [8] and other malaria endemic settings have also reported similarly large proportions of asymptomatic/sub-microscopic individuals using molecular diagnostic methods compared to affordable tools such as RDT or microscopy [29] (or placental impression smears [8]. This is very concerning since most of these peripheral malaria infections were asymptomatic and below the detection limit of conventional microscopy and/or rapid diagnostic tests and could, therefore, become an emergency [10] or evolve to severe disease if undetected or diagnosis was delayed. Pregnant women may become asymptomatic reservoirs with low density infections; remain unidentified during routine passive surveillance since *P. falciparum* sequestration in the placenta hinders their detection by RDT or microscopy, the currently available diagnostic methods often used by low-income countries [7, 30, 31]. The challenge of identifying malaria parasite-infected in parturient in the area and elsewhere, especially in the context of compounding low uptake of intervention program such as IPTp-SP, raises the need to consider incorporating molecular methods or other more cost-effective tools for better management of falciparum infection in this vulnerable group [6].

*P. falciparum*-parasitaemia below the threshold of microscopy but detected by PCR are likely to represent viable parasites at very low densities [32], since subpatent *P. falciparum* infections are also highly prevalent in nonpregnant adults and children [33, 34]. Nevertheless, it is also plausible that these may constitute phagocytized *P. falciparum* from exclusive placental malaria that cause positive amplification without viable parasites circulating [8, 32]. Such submicroscopic infections are thought to be relevant in sustaining transmission in low- or unstable transmission settings [35, 36] as well as the dry season in southwestern Cameroon.

The use of long-lasting insecticide treated nets and IPTp-SP remain the key interventions for the protection of pregnant women and their offspring against malaria and its attendant consequences [37]. Self-reported ITNs use of 65.4% by pregnant women is encouraging considering the reported efficacy of the control measure against placental parasitaemia, miscarriage/stillbirth and LBW [38]. The possibility of overestimating ITNs use due to self-reporting cannot be excluded as the participant may try to please the interviewer [10]. Nevertheless, although majority of the women reportedly took at least a dose of IPTp-SP at term in this study, only 1 in 2 (55.7%) used the recommended optimal doses ($\geq$ 3 doses) during pregnancy. This moderate uptake though better than previous reports of 20–26% use of $\geq$ 3 doses of IPTp-SP in the country [39] and Tanzania is still below the 60% target by the Ministry of Health [40]. Higher optimal IPTp-SP uptake was expected as reported recently in a hypoendemic area in Tanzania [12] since the main healthcare facility was located close to the households and parturient expected to frequently attend ANC and be sensitized about SP by well-trained health care personnel [41]. The suboptimal uptake and bednet usage may accrue to the late presentation to ANC care, since majority of the women were in the third trimester. It is also possible that the sociopolitical crisis affecting the area, in the last few years, limiting the free movement of persons and goods, may have impeded hospital attendance and accessibility to the chemoprevention, due to stock-outs of the drug. Scaled up of optimal IPTp-SP dosage and complementary interventions during this malaria holoendemic dry season as well as early malaria case detection and prompt treatment with effective anti-malarial drugs will be necessary to prevent severe malaria and deleterious clinical outcomes in pregnant women [12].

In line with previous reports of gravidity-dependent susceptibility to MIP in malaria endemic regions [42, 43], the risk of *P. falciparum* infection was highest in primigravidae and the proportion of submicroscopic parasitaemia among infected women decreased with increasing gravidity [31]. This suggests an improvement of immune protection during subsequent pregnancies, limiting *P. falciparum* not only to low parasite densities but even to submicroscopic levels [8]. Such low parasite densities, averaging 271 parasites per microliter of blood in this study, are expected since majority of the parturients were multigravidae and antibodies against specific placental tissue-adhering *P. falciparum* strains [44, 45] are thought to be gradually acquired with successive pregnancies [46].

In line with recent report among pregnant women in a low malaria transmission setting [12], the use of sub-optimal IPTp-SP doses was not associated with increased risk of malaria parasitaemia in this study. Although the lack of statistical difference between those who received optimal and sub-optimal doses for IPTp-SP suggests that the effect of the chemoprophylaxis may not be translated clinically [12], it is possible that our study was not powered to determine the effect of increasing dose on peripheral parasitaemia or that the time interval between the last administered IPTp dose and screening for parasite infection may be a stronger determinant for detecting parasitaemia rather than the cumulative number of doses a woman has received. This is essential to consider, particularly in areas where the daily likelihood for a woman to have infectious mosquito bites is higher.

Nonetheless, the use of IPTp was associated with protection against submicroscopic *P. falciparum* parasitaemia, in line with previous studies and reviewed evidence of reduced risk of

placental malaria in IPTp-SP users [47, 48]. This suggests that the control strategy is still effective and *P. falciparum* populations in the area remain susceptible to the SP despite the documented high prevalence of molecular markers against the drug.

In endemic countries, maternal malaria is thought to contribute to neonatal mortality indirectly through low birthweight and prematurity, though, to our knowledge, no strong evidence exists to support this association [49]. The relationship between submicroscopic *P. falciparum* and pregnancy outcomes remains contentious, and the present study is not the only one to report an apparent lack of association between submicroscopic peripheral infection and infant outcomes at birth [50, 51]. The fact that neither microscopic or submicroscopic *P. falciparum* infection detected at study inclusion were associated with adverse birth outcomes, suggests the anti-malarial treatments taken to manage infection, cleared most infections or reduced parasite densities, limiting deleterious downstream effects [29].

Gravidity was associated with significant reductions in birthweight. The reductions in birthweight in first pregnancies is in line with previous reports [27], translated into a high rate of LBW and preterm birth delivery in primigravid women, possibly due to their lack of pregnancy-specific immunity, or the lower general risk of LBW associated with increased gravidity. Further studies are required to establish the exact role of demographic, pre- and peri-natal risk factors on birth outcomes.

## Limitations

This was a semi- longitudinal study involving pregnant women followed up till delivery. Adherence on the use of IPTp-SP could not be assessed as the number of doses reported was based on self-reported SP dosage or documented in the ANC cards and plasma SP not measured to authenticating the information on the use of IPTp-SP among study participants. The fact that very few women had LBW infants and preterm delivery, potentially affected the power of the study to detect associations between infection and adverse outcomes. Additionally, some of the observed associations may be confounded by factors not documented in this study.

## Conclusion

Malaria still remains a significant public health concern, even in low transmission season, with conventional microscopy and rapid diagnostic tests unable to identify a large proportion of pregnant women with very low density parasitaemia. IPTp-SP seems to be relevant in controlling submicroscopic *P. falciparum* infections, which remains common in pregnant women, and are hard to diagnose, with potentially deleterious consequences for maternal and fetal health. Similar studies should be conducted in a hyperendemic setting with higher parasitaemia levels and to interrogate if IPTp induces protective responses to malaria.

## Acknowledgments

We thank the participants from the hospitals who made this study possible; and the Administrative and health personnel who assisted with this work.

## Author Contributions

**Conceptualization:** Tobias O. Apinjoh, Vincent N. Ntui, Vincent P. K. Titanji.

**Data curation:** Tobias O. Apinjoh, Vincent N. Ntui.

**Formal analysis:** Tobias O. Apinjoh.

**Funding acquisition:** Tobias O. Apinjoh.

**Investigation:** Vincent N. Ntui, Hanesh F. Chi, Marcel N. Moyeh, Cabrel T. Toussi, Joel M. Mayaba, Livinus N. Tangi, Pilate N. Kwi.

**Methodology:** Tobias O. Apinjoh, Vincent N. Ntui, Hanesh F. Chi, Cabrel T. Toussi.

**Project administration:** Tobias O. Apinjoh, Vincent N. Ntui, Hanesh F. Chi, Marcel N. Moyeh, Cabrel T. Toussi, Joel M. Mayaba, Livinus N. Tangi.

**Resources:** Tobias O. Apinjoh, Jodie Dionne-Odom, Alan T. N. Tita, Eric A. Achidi, Alfred Amambua-Ngwa.

**Supervision:** Tobias O. Apinjoh, Hanesh F. Chi, Judith K. Anchang-Kimbi, Alfred Amambua-Ngwa, Vincent P. K. Titanji.

**Validation:** Tobias O. Apinjoh, Vincent N. Ntui.

**Writing – original draft:** Tobias O. Apinjoh, Vincent N. Ntui.

**Writing – review & editing:** Tobias O. Apinjoh, Vincent N. Ntui, Marcel N. Moyeh, Jodie Dionne-Odom, Alan T. N. Tita, Eric A. Achidi, Vincent P. K. Titanji.

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
