## [Decision Letter · Decision Letter 0]

7 Jul 2022

PONE-D-21-41105Intermittent preventive treatment with Sulphadoxine-Pyrimethamine (IPTp-SP) is associated with protection against sub-microscopic P. falciparum infection in pregnant women during the low transmission dry season in southwestern CameroonPLOS ONE

Dear Dr. Opinjoh

Thank you for submitting your manuscript to PLOS ONE. After careful consideration, we feel that it has merit but does not fully meet PLOS ONE’s publication criteria as it currently stands. Therefore, we invite you to submit a revised version of the manuscript that addresses the points raised during the review process.

ACADEMIC EDITOR:All changes recommended by reviewers are mandatory to improve the quality of your manuscript.Please insert comments here and delete this placeholder text when finished. We recommend a major revision of your manuscript based on the PLOS ONE's publication criteria.Please submit your revised manuscript by July 30.best regardsjosé luiz vieira

We look forward to receiving your revised manuscript.

Kind regards,

José Luiz Fernandes Vieira

Academic Editor

PLOS ONE

2. Title in the uploaded manuscript exceeded 250 character limit. Please shorten the title to comply with journal requirements.

Reviewers' comments:

Reviewer's Responses to Questions

**Comments to the Author**

1. Is the manuscript technically sound, and do the data support the conclusions?

Reviewer #1: Yes

Reviewer #2: Partly

2. Has the statistical analysis been performed appropriately and rigorously? 

Reviewer #1: Yes

Reviewer #2: No

3. Have the authors made all data underlying the findings in their manuscript fully available?

Reviewer #1: Yes

Reviewer #2: Yes

4. Is the manuscript presented in an intelligible fashion and written in standard English?

Reviewer #1: Yes

Reviewer #2: Yes

5. Review Comments to the Author

Reviewer #1: Dear authors.

First, I want to congratulate your study. However, after analysis, some questions must be answered or justified before publication.

Their study deals with “Intermittent preventive treatment with Sulphadoxine-Pyrimethamine (IPTp-SP) is associated with protection against sub-microscopic P. falciparum infection in pregnant women during the low transmission dry season in southwestern Cameroon”.

I. Abstract section: includes all necessary information. II. Introduction: It includes all the necessary information, so you could mention other studies that were effective in this context and even the limitation in your state of the art. This way we will have more robustness in the study. III: Methodology: where was the material used for detection, on filter paper, for example, or was the material in natura? – this information must be described; in addition, I did not identify the process of sample feasibility, before carrying out the PCR process; as well, the criteria for admission of the drug to the patient, dosage and/or conflicting factors of the process were not described. However, it would be ideal to demonstrate the adverse reactions that these patients had after using the drug. IV: Results: includes all necessary information. V: Discussion: would you be able to correlate this process with the possible pharmacogenomic influence on patients, perhaps, this will give you the necessary answers from both positive and negative results. Finally, I congratulate you on your study and wish you success.

Reviewer #2: Comments:

1. Abstract:

Line 39: A sentence should not start with number.

No perspectives were drawn from this study in the conclusion section.

Line 53: May be country to be added here.

2. Introduction:

Line 85: Harrington et al. 2012. There is a lack of consistency in the reporting of reference, as some are in number. Same for reference in line 95, ect….

Line 96: the term effect is not adequate for this type of study. Association is preferable, as indicated in the title of the paper.

3. Method section:

In the ethic section, the suggestion is to provide the reference number of the ethical approval.

Line 121: reference 16 is cited. In fact this sentence is not from that reference when assessed in the reference list.

Line 144: reference to be numbered.

Line 157: when was haemoglobin assessed? At enrolment or at delivery?

Data analysis:

The analytic section is less developed than it should be. The authors should provide more details here.

Also,

Line 200: what were the criteria of having a variable in the multivariate model in tables that content adjustment? Also, it is not clear in this section that logistic regression has been used, although OR was seen in the tables.

Line 202: Chi Square test was used. Can the authors provide the Chi Square test used?

As this is cohort study, what is the rationale of using odds ratio instead of risk ratio?

4. Results:

In table 2: Fever at enrolment; What is the difference between the Febrile and Axillar temperature >= 37.5?

The prevalence of low birthweight seems to be very low. What is the authors’ impression on this, as in many study conducted in sub Saharan Africa this is about 10% ?

Line 252: the numerator reported here (35) is different from the sample size (57) to estimate the geometric mean of parasitemia. Can the authors explain that?

Line 255: The term chemoprophylaxis is used. However, the authors were assessing IPTp. These two terms are different. The term IPTp is preferred. Same for line 289.

Line 303: the term birth weight is used in two words. However, in many parts of the document, it is written in a single word.

Line 307: Similarly, to line 260, the term Effect is not appropriate, as this is an association study and not effect study.

Table 4: The authors seem to have no difference in term of birth weight in different groups of IPTp. Can the authors provide any explanation of that?

Table 4 and Figure 3: There seems to be a discrepancy between Table 4 and Figure 3 regarding the association between IPTp usage and dosage and malaria infection. Is there any explanation of that?

5. Discussion:

There is a mixture of Harvard and Vancouver across all the discussion section. Please correct.

6. Conclusion:

One would like to have at least a sentence for perspective in the context of this study. Can the authors provide one?

6. PLOS authors have the option to publish the peer review history of their article (what does this mean?). If published, this will include your full peer review and any attached files.

Reviewer #1: **Yes: **Luann Wendel Pereira de Sena

Reviewer #2: No

---

## [Author Response · Author response to Decision Letter 0]

13 Aug 2022

Reviewer Comment Response

Reviewer 2 

Abstract: 

Line 39: A sentence should not start with number. Corrected to start with ”About” 25%

No perspectives were drawn from this study in the conclusion section. Future studies should be carried out in hyperendemic malaria foci where the parasitemia levels are substantially higher in order to confirm the efficacy of IPTp-SP

Line 53: May be country to be added here. The country “Cameroon” where this study was undertaken has been added to the keywords as suggested

Introduction: 

Line 85: Harrington et al. 2012. There is a lack of consistency in the reporting of reference, as some are in number. Same for reference in line 95, ect…. All inconsistencies in citation and references have been corrected in line with journal specifications

Line 96: the term effect is not adequate for this type of study. Association is preferable, as indicated in the title of the paper. This has been corrected in the manuscript to “relationship” since pregnancy outcomes were not associated submicroscopic infection 

Method section: 

In the ethic section, the suggestion is to provide the reference number of the ethical approval. The reference number of the ethical approval (ref. No2019/09/1188/CE/CNERSH/SP) has now been provide as suggested

Line 121: reference 16 is cited. In fact this sentence is not from that reference when assessed in the reference list. Thanks for the correction and apologies for the mix-up. The correct reference (16 and not 15) has now been cited in the text.

Line 144: reference to be numbered. Corrected as suggested

Line 157: when was haemoglobin assessed? At enrolment or at delivery? Haemoglobin levels were assessed at enrolment and has now been indicated as such in the manuscript

Data analysis: 

The analytic section is less developed than it should be. The authors should provide more details here. This section has now been further developed and relevant details provided as recommended

Line 200: what were the criteria of having a variable in the multivariate model in tables that content adjustment? Also, it is not clear in this section that logistic regression has been used, although OR was seen in the tables. The use of logistic regression and criteria for including variables in multivariate models has now been detailed in the analysis section 

Line 202: Chi Square test was used. Can the authors provide the Chi Square test used?

As this is cohort study, what is the rationale of using odds ratio instead of risk ratio? The corresponding chisquare statistics has now been incorporated into the table as suggested. The analysis in question relates to the risk of the participants at one time point (enrolment) and not following the semi-longitudinal outcome

Results: 

In table 2: Fever at enrolment; What is the difference between the Febrile and Axillar temperature >= 37.5? Fever at enrolment was classified in different ways. Current fever was defined as body temperature ≥37.5°C, with febrile individuals defined as participants with current fever or history of fever at enrolment. This has now been clarified under the “Data Collection and Definition of terms’ section in the manuscript

The prevalence of low birthweight seems to be very low. What is the authors’ impression on this, as in many study conducted in sub Saharan Africa this is about 10%? 

 This may be due to multiple factors including the status, nutrition and adherence of the mothers to control and intervention strategies. We note that most of the women were aged > 25 years, multigravidae, IPTp-SP users of atleast 3 doses at term, which factors have been shown to contribute to improved pregnancy outcome. 

Line 252: the numerator reported here (35) is different from the sample size (57) to estimate the geometric mean of parasitemia. Can the authors explain that? This can be explained by the fact that a few samples assessed for P. falciparum infection by peripheral blood smear microscopy were not assessed by PCR and vice versa. Such discrepancy would accrue to damaged slides during transportation/processing or indeterminate PCR results.

Line 255: The term chemoprophylaxis is used. However, the authors were assessing IPTp. These two terms are different. The term IPTp is preferred. Same for line 289. This has been corrected in the manuscript as suggested

Line 303: the term birth weight is used in two words. However, in many parts of the document, it is written in a single word. This has been corrected in the manuscript as suggested 

Line 307: Similarly, to line 260, the term Effect is not appropriate, as this is an association study and not effect study. This has been corrected in the manuscript as suggested

Table 4: The authors seem to have no difference in term of birth weight in different groups of IPTp. Can the authors provide any explanation of that? 

 Indeed. Although there was a trend in creasing birthweight with IPTp-SP use, there was no difference in different IPTp-SP groups. This suggest that although the chemotherapy is associated with protection from sub-microscopic parasitaemia, this does not translate into pregnancy outcome in terms of birthweight. It is possible that the low numbers of non IPTp-SP users at term may not be statistically powered to detect difference, if they do exist. 

Table 4 and Figure 3: There seems to be a discrepancy between Table 4 and Figure 3 regarding the association between IPTp usage and dosage and malaria infection. Is there any explanation of that? 

 Table 4 and Figure 3 present completely different information vis-à-vis IPTp usage and dosage.

While Table 4 compares pregnancy outcome (birthweight and APGAR score) across different IPTp usage and dosage groups, Figure 3 shows the association between IPTp-SP usage and dosage and the malaria parasite infection status

Discussion: 

There is a mixture of Harvard and Vancouver across all the discussion section. Please correct. All inconsistencies in citation and references have been corrected in line with journal specifications

Conclusion: 

One would like to have at least a sentence for perspective in the context of this study. Can the authors provide one? To conduct similar studies in a hyperendemic setting with higher parasitaemia levels and interrogate if IPTp induces protective responses to malaria

Reviewer Comment Response

Reviewer 1 

I. Abstract section 

includes all necessary information All necessary information has been included 

II. Introduction 

It includes all the necessary information, so you could mention other studies that were effective in this context and even the limitation in your state of the art. This way we will have more robustness in the study. All necessary information has been included

III: Methodology: 

where was the material used for detection, on filter paper, for example, or was the material in natura? – this information must be described; in addition, I did not identify the process of sample feasibility, before carrying out the PCR process; as well. This has been corrected in the manuscript to include details on filter paper type and sample storage conditions prior to PCR testing. The study describes status in natura as testing was done on samples directly collected from the women enrolled.

the criteria for admission of the drug to the patient, dosage and/or conflicting factors of the process were not described. However, it would be ideal to demonstrate the adverse reactions that these patients had after using the drug. A subsection detailing IPTp-SP usage and dosage information has now been included in the methodology section. Note that the previous drug use was self-reported or documented from medical records and subsequent use captured by directly observed administration by health care personnel.

IV: Results: 

includes all necessary information. All necessary information has been included

V: Discussion 

would you be able to correlate this process with the possible pharmacogenomic influence on patients, perhaps, this will give you the necessary answers from both positive and negative results. Finally, I congratulate you on your study and wish you success. It would be interesting to correlate out observation with pharmacogenomics influence on participants, a perspective in view of future studies. Thank you for the brilliant suggestion.

---

## [Decision Letter · Decision Letter 1]

16 Sep 2022

Intermittent preventive treatment with Sulphadoxine-Pyrimethamine (IPTp-SP) is associated with protection against sub-microscopic P. falciparum infection in pregnant women during the low transmission dry season in southwestern Cameroon

PONE-D-21-41105R1

Dear Dr.Tobias Apinjoh

We’re pleased to inform you that your manuscript has been judged scientifically suitable for publication and will be formally accepted for publication once it meets all outstanding technical requirements.

Kind regards,

José Luiz Fernandes Vieira

Academic Editor

PLOS ONE

Additional Editor Comments (optional):

Reviewers' comments:

Reviewer's Responses to Questions

**Comments to the Author**

1. If the authors have adequately addressed your comments raised in a previous round of review and you feel that this manuscript is now acceptable for publication, you may indicate that here to bypass the “Comments to the Author” section, enter your conflict of interest statement in the “Confidential to Editor” section, and submit your "Accept" recommendation.

Reviewer #1: All comments have been addressed

2. Is the manuscript technically sound, and do the data support the conclusions?

Reviewer #1: Yes

3. Has the statistical analysis been performed appropriately and rigorously? 

Reviewer #1: Yes

4. Have the authors made all data underlying the findings in their manuscript fully available?

Reviewer #1: Yes

5. Is the manuscript presented in an intelligible fashion and written in standard English?

Reviewer #1: Yes

6. Review Comments to the Author

Reviewer #1: Dear,

Congratulations again for the study. All suggestions are fulfilled. Through the suggestions the text became more solid and understood.

7. PLOS authors have the option to publish the peer review history of their article (what does this mean?). If published, this will include your full peer review and any attached files.

Reviewer #1: No

---

## [Editor Report · Acceptance letter]

22 Sep 2022

PONE-D-21-41105R1 

Intermittent preventive treatment with Sulphadoxine-Pyrimethamine (IPTp-SP) is associated with protection against sub-microscopic P. falciparum infection in pregnant women during the low transmission dry season in southwestern Cameroon 

Dear Dr. Apinjoh:

I'm pleased to inform you that your manuscript has been deemed suitable for publication in PLOS ONE. Congratulations! Your manuscript is now with our production department. 

Kind regards, 

on behalf of

Dr. José Luiz Fernandes Vieira 

Academic Editor

PLOS ONE